# *Limosilactobacillus reuteri* in Health and Disease

**DOI:** 10.3390/microorganisms10030522

**Published:** 2022-02-28

**Authors:** Jumana Abuqwider, Mohammad Altamimi, Gianluigi Mauriello

**Affiliations:** 1Department of Agricultural Science, University of Naples Federico II, 80049 Naples, Italy; jqwider@gmail.com; 2Department of Nutrition and Food Technology, Faculty of Agriculture and Veterinary Medicine, An-Najah National University, Nablus P.O. Box 7, Palestine; m.altamimi@najah.edu

**Keywords:** obesity, glucose homeostasis, hepatic disease

## Abstract

*Limosilactobacillus reuteri* is a microorganism with valuable probiotic qualities that has been widely employed in humans to promote health. It is a well-studied probiotic bacterium that exerts beneficial health effects due to several metabolic mechanisms that enhance the production of anti-inflammatory cytochines and modulate the gut microbiota by the production of antimicrobial molecules, including reuterin. This review provides an overview of the data that support the role of probiotic properties, and the antimicrobial and immunomodulatory effects of some *L. reuteri* strains in relation to their metabolite production profile on the amelioration of many diseases and disorders. Although the results discussed in this paper are strain dependent, they show that *L. reuteri*, by different mechanisms and various metabolites, may control body weight and obesity, improve insulin sensitivity and glucose homeostasis, increase gut integrity and immunomodulation, and attenuate hepatic disorders. Gut microbiota modulation by ingesting probiotic *L. reuteri* strains could be a promising preventative and therapeutic approach against many diseases and disorders.

## 1. Introduction

The use of randomized, blind, or double-blind human trials has fueled a surge in interest in the field of probiotics in recent years, coinciding with a renewed interest in studies concentrating on gut microbial ecology [1]. Probiotics are defined as “live microorganisms which when administered in adequate amounts, confer a health benefit on the host” by the World Health Organization [2]. The *Lactobacillus* and *Bifidobacterium* strains are the microorganisms most used for food probiotication and supplements, and a growing number of new strains are achieving the role of probiotics [3]. Prebiotics are closely associated with probiotics, and refer to compounds that are resistant to digestion but fermentable in the gut and able to stimulate the growth of probiotics [4].

Lactobacilli have been utilized for a long time in fermentation processes to preserve foods [5]. Lactobacilli’s main antibacterial activity is due to the release of lactic acid, which lowers the pH of the surrounding environment and the internal cell pH of pathogens. However, lactic acid is not the only organic acid involved in the antibacterial activity. Indeed, lactobacilli can produce other organic acids, such as acetic, propionic and phenyl lactic acids, which contribute both to the drop in pH and the potential inhibition of the growth of pathogenic microorganisms. Moreover, lactobacilli produce a wide variety of antimicrobial molecules, including low-molecular-mass compounds such as hydrogen peroxide, carbon dioxide, ethanol, diacetyl, and acetaldehyde, as well as more complex molecules like bacteriocins, reuterin and reutericyclin, which are the final products of metabolism performed by *Limosilactobacillus reuteri* strains [6,7,8,9]. Apart from their antibacterial capabilities, lactobacilli have been shown to interact with the host immune system, impacting mucosal immune cells and epithelial cells that coat the mucosa to activate the mucosal immune system’s processes. The Toll-like receptors (TLRs), the nucleotide-binding oligomerization domain-like receptor, and C-type lectin receptors are the three main signaling systems that allow the innate immune system to recognize lactobacilli. Lactobacilli can have an effect on the immune response of the host by different actions, including the cell-surface carbohydrates, the enzymes that modify the structure of lipoteichoic acids and various metabolites [10]. The biogenic amines, histamine and tyramine in particular, are metabolites from lactobacilli that can affect the immune system of the host [11]. However, the production of antibacterial and immunomodulatory chemicals is highly dependent on the strain and growth state [12].

*L. reuteri* is a well-studied probiotic bacterium with the ability to colonize a wide range of animals. It can be found in various parts of the human body, including the gastrointestinal (GI) tract, urinary tract, skin, and breast milk [13]. It is one of the few ex-*Lactobacillus* species specifically adapted to survive in the GI tract as it occurs naturally in the human intestine [14].

When *L. reuteri* is ingested, the protective effects are reported to be due to different metabolic mechanisms. First, certain strains of *L. reuteri* produce an anti-inflammatory compound that lowers the expression of the proinflammatory cytokine tumor necrosis factor alpha (TNF-α) in cultured activated macrophages by more than 90% [15,16]. Second, *L. reuteri* produces reuterin, an antibacterial molecule that is bactericidal against a wide range of species, including enterohemorrhagic *Escherichia coli* strains [17]. Reuterin, also known as 3-hydroxypropionaldehyde [3-HPA], is an intermediate in the metabolism of glycerol to 1,3-propanediol, guaranteeing that the cell replenishes NAD+ during glucose metabolism [18]. This review provides an overview of data that support the role of probiotic properties, and the antimicrobial and immunomodulatory effects of *L. reuteri* strains in relation to their metabolism on the amelioration of many diseases and disorders (Figure 1).

## 2. *Limosilactobacillus reuteri* and Obesity Control

Obesity is a disease that is characterized by a pathological accumulation of body fat accompanied by a decline in the state of health and quality of life. The correlation between obesity and the Western diet is well established [19]. Obesity is a major public health problem worldwide and it has a decisive impact on the duration of life because it can lead to the onset of high blood pressure, diabetes mellitus, sleep apnea and cardiovascular disease. Nowadays, the strong correlation between the gut microbiota and obesity is well established [20]. In particular, alterations in the abundance of *Bacteroidetes* and *Firmicutes*, as significant bacterial divisions, have been connected to changes in the gut microbiota and its role in obesity (Figure 2).

The therapeutic significance of probiotics is in managing gut dysbiosis, which is characterized by an aberrant microbiota composition, as well as decreased permeability of the intestinal barrier and inflammatory activation [21]. Notably, *Lactobacillus* and *Bifidobacterium* has been examined to identify their capacity to balance intestinal microbiota and for obesity treatment [22,23].

Here, we want to clarify that the term microbiota means the live organisms of an ecosystem, so the gut microbiota is a complex community of live microorganisms in the gut. Everyone has an individualized gut microbiota profile, which is not stable during their lifetime. So, there is no implication of health in the term itself, but probiotics can help to balance the gut microbiota so that is has a healthier composition [24].

*L. reuteri* is one of the species investigated for their effect on obesity and conflicting findings are reported in the current literature. Indeed, some authors have shown the enrichment of gut microbiota by *L. reuteri* with a depletion in *Bifidobacterium animalis* and *Methanobrevibacter smithii* by analyzing the stools of 68 obese people and 47 controls [25]. On the other hand, according to another study, the *L. reuteri* MG5149 strain reduced body weight, adipose tissue weight, and adipocyte size in epididymal tissue. In addition, it inhibited the expression of lipogenic proteins such as peroxisome proliferator-activated receptor, CCAAT/enhancer-binding protein, fatty acid synthase, and adipocyte-protein 2 in some tissues and reduced fat accumulation by upregulating the phosphorylation of AMP-activated protein kinase and acetyl-CoA carboxylase [26]. Furthermore, treatment with *L. reuteri* JBD30l lowered the concentration of free fatty acids (FFAs) in the small intestine’s gut fluid, lowering intestinal FFA absorption while increasing fecal FFA excretion [27]. However, there are contradictory results indicating that *L. reuteri* may stimulate the progress of obesity in a strain-dependent way. The link was confirmed when vancomycin-resistant *L. reuteri* in the gut microbiota was discovered to be a predictor of body weight increase following vancomycin treatment [28]. Other studies have associated weight gain with *L. reuteri* abundance, due to the level of fructose ingested [29]. *L. reuteri* may use fructose molecules as a source of energy to increase its growth rate, resulting in the high absorption of fructose and increasing the synthesis of intermediary molecules to produce triglycerides [29].

The regulation of energy homeostasis and satiety, alteration of gut microbiota composition, synthesis of short chain fatty acids (SCFA), improved gut barrier function, and the interruption of bile acid metabolism in the host are all the proposed mechanisms of action for probiotic-mediated weight loss [30]. As a matter of fact, *L. reuteri* is one of the probiotic bacteria that can perform these mechanisms to control obesity (Figure 2). On the other hand, food ingredients that enhance the growth of *L. reuteri* in the gut result in weight loss [31]. Further research is needed to determine whether an increase in *L. reuteri* is the cause of weight reduction per se or whether it is substrate-dependent.

## 3. *Limosilactobacillus reuteri* and Improvement in Insulin Sensitivity and Glucose Homeostasis

Type 2 diabetes mellitus (T2DM) is characterized by hyperglycemia and insulin insensitivity associated with a diminished incretin response, subclinical inflammatory processes, and impaired glucose tolerance [32]. The insulin resistance and the postprandial glucose response can be regulated by the modulation of the gut microbiota in animal models [33]. Furthermore, the enhanced intestinal barrier function is linked to the lowering of portal lipopolysaccharide (LPS) endotoxin concentrations and of systemic and hepatic inflammation [34]. Simon et al. [32] demonstrated that daily treatment of *L. reuteri* SD5865 increased glucose-stimulated GLP-1 and GLP-2, improved insulin sensitivity, and elevated insulin secretion through increasing incretin release. The blood glucose-lowering action of GLP-1 is terminated due to its enzymatic degradation by dipeptidyl-peptidase-IV (DPP-IV) [35]. Another study showed that oral gavage of *L. reuteri* GMNL-263 (Lr263) decreased serum glucose, insulin resistance, leptin, C-peptide, and GLP-1. Moreover, when Lr263 was given, the level of HbA1c, a key indicator of glucose metabolism, was reduced [36]. The consumption of Lr263 dramatically decreased the load of *Clostridia* and *Bacteroidetes* while boosting the number of bifidobacteria and lactobacilli. These findings suggested that Lr263 treatment may have a therapeutic effect on diabetes, particularly by increasing *Bifidobacterium* spp. [36,37]. However, Mobini et al. [38] did not find any improvement in HbA1c in people with T2DM on insulin therapy after 12 weeks of *L. reuteri* DMS intake. A subset group of patients benefited from the treatment, which may be due to the diversity in their gut microbiota.

The reduction in pro-inflammatory cytokines, enhanced bile salt hydrolase (BSH) activity, and alterations in intestinal microbiota composition are all probable explanations for the positive effects of *L. reuteri* consumption on T2DM treatment [36]. It has been suggested that taking probiotics can increase the integrity of the intestinal epithelium and limit the TLR 4 pathway, thus lowering pro-inflammatory signals and improving insulin sensitivity [39,40]. Microbial BSH activity has been demonstrated to increase the level of unconjugated bile acids and activate G protein-coupled receptor 5 (TGR5) to improve insulin sensitivity [41].

## 4. *Limosilactobacillus reuteri* and Immunomodulation of the Gut Microbiota

Changes in the intestinal microbiota have been identified as a key factor in the increase in the occurrence of certain GI disorders [42]. However, in recent years, the efficacy of probiotics in the prevention and treatment of GI diseases has received a lot of attention [43,44]. As mentioned before, probiotics are strain-specific in their efficiency, and each strain may contribute to host health through distinct mechanisms. Three major metabolic strategies are utilized by probiotic microorganisms, which provide actions for modulating the gut microbiota. The first is competition for nutrients and space with other microorganisms, which slows the growth of hazardous bacteria, thus limiting their pathogenic potential [45]. They can also have a direct influence on other bacteria, which inhibits pathogen adherence and colonization of the GI tract [46]. The second approach implicates the production of antimicrobial compounds such as bacteriocins, organic acids, and short-chain fatty acids [47]. Probiotics also inhibit the growth of potentially hazardous bacteria in the GI tract by changing the environment to an unfavorable one, causing pathogens to die-off and toxin inactivation [48]. The third function of probiotics is the stimulation and control of the immune system, including specific and non-specific reactions. This is accomplished through T lymphocyte activation, cytokine generation, phagocytosis induction, and stimulation of IgA antibody secretion [49,50]. Probiotics induce the body to produce defensins, antimicrobial proteins, and lecithin through interacting with many immune system cells. Consequently, the body is benefited by changing the overall profile of pro- and anti-inflammatory cytokines, and enabling the inactivation of potentially harmful microbes and/or their toxins [51,52].

The impact of modifying the microbiota of the small intestine is considered in modern theories regarding functional disorders of the GI tract [53]. Dysbiosis is characterized by an aberrant microbiota composition, as well as decreased permeability of the intestinal barrier and inflammatory activation [21]. *L. reuteri* DSM 17938 is one of the most thoroughly investigated probiotics in children and adults with functional GI problems [54]. The ability of a probiotic strain to adhere to the human GI tract is critical for colonization, interaction with host cells, pathogen suppression, and epithelial cell protection or immunological modulation [55]. Several studies have documented *L. reuteri*‘s ability to colonize, and its capacity to adhere to mucin and intestinal epithelial cells. The probable mechanism involved in adhesion has been linked to the surface protein, mucus-binding protein [56], antiadhesive properties of bacterial exopolysaccharides (reuteran and levan) [57], inulosucrase enzyme [58], D-alanyl-LTA [59], and glucosyltransferase A [54].

Furthermore, one of the best-documented probiotic pathogen-inhibiting methods is *L. reuteri*’s antimicrobial activity. Antimicrobial compounds produced by *L. reuteri* include lactic acid, acetic acid, ethanol, reuterin [60,61,62], reutericyclin [63], and the anti-inflammatory molecule, histamine [64].

Reuterin is a mixture of different forms of 3-hydroxypropionaldehyde (3-HPA) [65]. It is well known that *L. reuteri* can metabolize glycerol to generate 3-HPA in a coenzyme B12-dependent, glycerol dehydratase-mediated reaction [66]. Furthermore, the strength of antimicrobial activity appears to be correlated to the natural pH-dependent interconversion of 3-HPA to acrolein and vice versa [67]. Acrolein is a highly reactive and electrophilic α,β-unsaturated aldehyde, and on the basis of its structure, is anticipated to be more reactive than 3-HPA with regards to nucleophile addition. Reuterin has the ability to conjugate heterocyclic amines in addition to its antibacterial properties, which appears to be strongly dependent on the synthesis of acrolein [68], suggesting it has an essential role in the antimicrobial activity of *L. reuteri* strains.

Lactic acid, acetic acid, ethanol, and reutericyclin have all been found as products of various *L. reuteri* strains, as previously mentioned. *L. reuteri* has been proven to be effective against a variety of bacterial infections in the GI tract thanks to the synthesis of these compounds. *E. coli*, *Clostridium difficile*, and *Salmonella* are included in these infections [69,70,71]. The use of *L. reuteri* to treat *Helicobacter pylori* is one of the most well-known examples of *L. reuteri’s* efficacy as a probiotic against infections. Infection with *H.*
*pylori* is a leading cause of chronic gastritis and peptic ulcers, as well as a possible risk for stomach cancer [72]. It has been reported that *L. reuteri* acts by contending with *H. pylori* for glycolipid receptors binding, then suppressing its growth. Consequently, *H. pylori* bacterial burden is reduced and the associated symptoms are also minimized [73].

A few strains of *L. reuteri* may convert the amino acid L-histidine, which is found in many foods, to the biogenic amine histamine [60]. Thomas et al. [74] reported that histamine produced from *L. reuteri* 6475 reduced the production of tumor necrosis factor (TNF). The activation of the histamine H2 receptor, which increased intracellular cAMP and protein kinase A, and the inhibition of MEK/ERK signaling were all required for this suppression [74]. The findings of this research point to the possible use of *L. reuteri* in the treatment of people with GI abnormalities. Similarly, osteoporosis that is predominant in Type 1 diabetes, which is mediated by TNF-α and suppression of Wnt10b expression, was prevented in animal model by *L. reuteri* [75].

## 5. *Limosilactobacillus reuteri* Attenuates Hepatic Disorders

The liver is constantly exposed to substances derived from the gut, such as microbial components and their products [76,77]. Hepatocytes express pattern recognition receptors, which identify bacterial components such as LPS and activate inflammatory pathways [78]. Growing evidence has revealed that a weak gut barrier, also referred to as “leaky gut”, enhances the interaction between gut bacteria and hepatic receptors, such as Toll-like receptors (TLR) [79]. In addition, TLR signals have a crucial role in the regulation of the innate immune response in the liver [77]. LPS, for instance, stimulates TLR4 in Kupffer cells, causing pro-inflammatory cytokines like TNF-α to be produced, causing hepatocyte injury. As a result, the inflammatory response caused by the gut–liver axis imbalance leads to the onset and progression of liver disorders [80], such as liver cirrhosis [81], non-alcoholic fatty liver diseases [82] and hepatic encephalopathy [83]. On the other hand, changes in the gut microbiota with probiotics have been shown to have a potential positive influence on hepatic disorders [84] (Figure 3). Thus, a treatment based on microbial intervention is a promising method to improve the pathological process in the liver [85].

Probiotics, particularly *Lactobacillus*, can limit the growth of pathogens while also improving the host’s immunological and metabolic functioning. *L. reuteri* strains are widespread in the mammalian gut and have health-promoting qualities [13]. Jiang et al. [86] showed that treatment in advance with *L. reuteri* DSM 17938 reduced gamma-glutamyl transferase, serum alanine aminotransferase, aspartate aminotransferase, IL-1, IL-2, IL-18, M-CSF, and MIP-3 levels, as well as tissue disorders in the last part of the ileum and liver. Additionally, *L. reuteri* DSM 17938 decreased the abundance of some potential pathogenic taxa, such as *Actinomycetales*, *Coriobacteriaceae, Staphylococcaceae*, and *Enterococcaceae*. Furthermore, it lowered the production of inflammatory genes in the liver, such as Ccl2, Ccl7, Ccl11, Ccl12, Il6, Il11, Il20rb, Mmp3 and Mmp10 [87].

Additionally, *L. reuteri* DSM 17938 relieves liver failure in several ways including the downregulation of retinol metabolism and the peroxisome proliferator-activated receptor (PPAR) signaling pathway, as well as the elevation of viral protein association with cytokine and cytokine receptors, and the central carbon metabolism in the cancer signaling pathways [36,88]. Previous animal studies have demonstrated that the administration of *L. reuteri* 263 significantly lowered aspartate aminotransferase (AST). Furthermore, the alanine aminotransferase (ALT) levels and the antioxidant activities of superoxide dismutase and glutathione reductase in rat livers were decreased [89]. Another study found that probiotic *L. reuteri* GMNL-89 and *L. reuteri* GMNL-263 reduced hepatic inflammation and apoptosis in lupus-prone rats by decreasing the MAPK and NF-B signaling pathways [90]. The preservation of normal epithelial cell structure and function, particularly tight junction (TJ) proteins, prevents bacteria from migrating transepithelially or paracellularly. According to some studies, short-chain fatty acids, proteins, polyamines, and proteins p75 and p40 are considered to have a critical role in providing beneficial effects on gut dysbiosis and liver disease through augmentation of the immune function and gut barrier integrity [91,92,93,94].

Finally, Cui et al. [80] indicated that *L. reuteri* ZJ617s diminished gut permeability and suppressed the secretion of hepatic pro-inflammatory cytokines by inhibiting the TLR4/MAPK/NF-B and autophagy signaling pathways, which decreased plasma concentrations of AST and ALT, and alleviated hepatic histological changes.

## 6. *Limosilactobacillus reuteri* and the Gut–Brain Axis

It is well established that brain–gut communications and vice versa have pathophysiological consequences that range from altered responses to chronic inflammation to behavioral states [95]. Microbiota play a key role in this bidirectional communication, making it a target for much research, especially in the development of various neuropsychiatric conditions [96].

Psychological stress and inflammation are common threads in the pathophysiology of disorders in which microbiota may have a role. Depression, schizophrenia, autism spectrum disorder (ASD), epilepsy, and migraine are all caused by stress, whereas depression, schizophrenia, ASD, Parkinson’s disease, epilepsy, and migraine are all caused by inflammation [97].

Research that addresses the impact of microbiota on the gut–brain axis has used complementary medicine, germ-free mice, antibiotic-treated animal models, supplementation with probiotics and fecal transplantation [95,96]. Neurogenesis, a critical process in learning and memory was reported to be regulated by microbiome [96]. On the other hand, stress and emotions will shape the gut microbiome through hormones and neurotransmitters. Neurotransmitters such as acetylcholine catecholamines, GABA and serotonin are produced by microbiota *Lactobacillus*, *Bifidobacteria*, *Enterococcus* and *Streptococcus* spp. in the gut, which greatly affect the brain function [96,98]. Moreover, serotonin, an internal mood tranquilizer, is 90% produced in the GI tract [99]. A major microbiota metabolite, SCFA, was found to modulate microglial cells’ development and function [100]. It was shown that the intake of probiotics/prebiotics may modulate brain function and behavior [100]. Patients with psychiatric disorders have shown different microbiota profiles compared to healthy individuals, with a tendency to have lower diversity and an imbalance of bacteria, towards harmful bacteria. Psychobiotics are probiotics that target one or more of the brain functions or disorders such as depression, anxiety, schizophrenia and autism through microbiota–brain interactions [101]. Although there are inconsistencies in the outcomes of treatment with psychobiotics due to dose, strains and duration of treatment by probiotics, such treatments are promising, with no side effects compared to other drugs [102]. ASD is a disorder with no medication to treat the core symptoms; however, Santocchi et al. [103] found that improvement in core symptoms of ASD by the use of probiotics was related to improvement in GI dysfunction, confirming the connection between the gut–brain axis and microbiota. On the other hand, *L. reuteri* showed an improvement in reversing the social deficit in genetic, environmental, and idiopathic ASD models [104]. In a pilot study of oral administration of 10^10^ colony forming units of *L. reuteri* daily or placebo for a duration of 3 months to ASD children, the authors proposed that the mechanistic understanding of *L. reuteri* was based on the gut–brain axis, where *L. reuteri* altered the GI function and gut microbiome [105]. In a genetic animal model of autism, a decreased level of *Lactobacillus* spp. was noted. The abundance of *L. reuteri* correlated significantly with the expression of each of the three GABA receptor subunits and attenuated the unsocial behavior of animals [106]. In the latter study, the authors concluded that treatment with *L. reuteri* regulated the behavior and several molecular mechanisms associated with ASD.

## 7. *Limosilactobacillus reuteri* and the Management of Inflammatory Bowel Disease

Chronic gut inflammation of uncertain etiology characterizes inflammatory bowel disease (IBD), which comprises ulcerative colitis (UC) and Crohn’s disease (CD). IBD’s main clinical symptoms include abdominal discomfort, diarrhea, and hematochezia, all of which have a negative impact on patients’ quality of life [107]. Abnormal immune responses, genetic predisposition, intestinal dysbiosis, recurrent intestinal infections, chronic intestinal mucosal barrier injury, poor nutrition, and other factors all have a role in the onset and progression of IBD [108]. There is a strong relationship between the composition of the intestinal microbiota and the incidence of IBD, according to several sources [109]. Patients with IBD have much less diversity in their gut microbiota than healthy people, according to a sequencing-based comparison. Evidently, the number of hazardous bacteria such as Bacteroides and Enterobacteria (including *E. coli*) increases, whereas the proportion of beneficial Firmicutes declines [110,111]. Additionally, in patients with IBD, a decrease in the number of bacteria that produce SCFAs has been found [112]. *L. reuteri*, when administered as a probiotic, helped to restore the balance of the gut microbiota [13]. In addition to the previously mentioned protective mechanisms for pathological intestinal microbiota disorder and immune dysregulation, recent mechanistic investigations revealed that under certain culture circumstances, *L. reuteri* CCM 3625 produces biogenic amines, such as histamine and tyramine, which may decrease the inflammatory response in the gastrointestinal tract [60]. It has been reported in one study that treatment with *L. reuteri* R2LC or 4659 decreased inflammation in the intestinal mucosa in a mouse model, acting by reducing the expression of proinflammatory markers [113]. Another recent study found that administration of *L. reuteri* F-9-35 led to anti-inflammatory benefits in a mouse model of colitis [114]. The reduced transcription of mRNA for COX-2, TNF-α, and IL-6, as well as the restoration of the intestinal equilibrium between *Firmicutes* and *Bacteroidetes*, were credited with this protective effect [114].

Innate immune responses are provoked by the recognition of bacterial pathogen-associated molecular patterns (PAMPs) by the host pattern recognition receptors (PRRs) present on leukocytes, including Toll-like receptors (TLRs), NOD-like receptors (NLR), and C-type lectin receptors (CLRs), which form the proinflammatory response that is thought to be the pathogenic basis of IBD [115]. Studies of animal models with reduced expression and activation of NOD-like receptor or TLR signaling have shown the complex and context-dependent effect of innate immunity in colitis, ranging from preventive to proinflammatory [116]. According to a recent study, *L. reuteri* DSM 17938 reduced experimental necrotizing enterocolitis by inducing tolerogenic intestinal dendritic cells (DCs) and Tregs, which in turn decreased the proliferation of proinflammatory lymphocytes and activation of inflammatory cytokines via a process involving TLR2 [117].

Tryptophan (Trp) is an anti-inflammatory essential amino acid that supports the intestinal microbiota. Interestingly, its concentration in the intestinal lumen may be related to *Lactobacillus*-mediated regulation of intestinal immunity [107]. Cervantes-Barragan et al. [118] were the first to show that *L. reuteri* promotes the differentiation of T cells into CD4+CD8αα+ double-positive intraepithelial T lymphocytes (DPIELs) by metabolizing Trp to indole-3-lactic acid, which then activates the aryl hydrocarbon receptor (AhR) on CD4+ T cells to downregulate the transcription factor Thpok and ultimately induce their differentiation into DPIELs. A study showed that some strains of *L. reuteri* activate AhR via the polyketone synthase cluster (PKS). Activation of AhR is required for the production of interleukin-22 (IL-22), which can boost the innate immune response by inducing the production of antimicrobial peptides (Reg3-lectins) to combat intestinal pathogens and tight junction proteins to protect intestinal tissues from inflammation damage [119]. Other studies have found that *L. reuteri* 5289 causes dendritic cells (DCs) to release IL-10 and inhibits the production of IL-12 by DCs in response to co-culture with other bacteria that typically induce IL-12 production. Interestingly, the *L. reuteri*-mediated inhibition of IL-12 production was linked to prolonged ERK1/2 MAP kinase phosphorylation [120].

## 8. *Limosilactobacillus reuteri* and Cystic Fibrosis

Cystic fibrosis (CF) is the most common autosomal recessive disease in the Caucasian population. Even though many initiatives have been undertaken to establish the prevalence of this illness, a number of reasons makes it difficult to ascertain the exact number of cases, including the wide range of quality in the medical/scientific literature and patient registries. However, most sources suggest that the disease affects 1 in 2500 to 3500 of the Caucasian population. CF is a multi-organ disease, which mainly affects the respiratory and digestive systems. A mutated gene, called the Cystic Fibrosis Transmembrane Regulator (CFTR) gene, causes the production of dehydrated and viscous mucus. This mucus overproduction leads to lung disfunction, closes the bronchial tubes, causes repeated respiratory infections, obstructs the pancreas and prevents pancreatic enzymes from reaching the intestine, consequently food cannot be correctly digested and assimilated [121,122]. Moreover, the inefficiency of the pancreas leads to a decline in intestinal motility, overgrowth of the bacterial population in the small intestine, functional gastrointestinal symptoms and intestinal obstruction [123]. The importance of the gut microbiota in health and disease is generally recognized, and it is increasingly recognized as an essential contributor to immunological and metabolic balance [124]. Dysbiosis, or disruption of the gut microbiota has been linked to CF gut patients with an excess of potentially harmful bacteria and a decrease in beneficial bacteria [125]. The administration of probiotics is increasingly being used to modulate the gut microbiota as a therapeutic technique for preserving health and treating diseases. Probiotics may help to repair the intestinal microbial balance disrupted by the regular use of antibiotic medications for the treatment and prevention of respiratory exacerbations in people with cystic fibrosis [126]. Some strains of *L. reuteri* have been reported to exert immunoregulatory properties and improve the gut health of CF patients. Di Nardo et al. [127] reported that *L. reuteri* ATCC55730 was found to be helpful in lowering the incidence of pulmonary exacerbations and the number of upper respiratory tract infections in patients with cystic fibrosis [127]. Although there is a long-held belief that probiotics can benefit by decreasing intestinal permeability, it is now commonly accepted that probiotics’ principal intestinal and extraintestinal effects are mediated through their interaction with gut immunity [128,129]. In this context, some authors have hypothesized a gut–lung axis of probiotic activity, resulting in improvement in both the innate and adaptive immune defenses in the respiratory tract because of the stimulus from probiotic microorganisms in contact with the gut-associated lymphoid tissue [127]. The mechanism involved in this gut–lung axis could be the increase in the IgA-secretory cells in the bronchial mucosa; the activation of natural killer cells, also known as NK cells or large granular lymphocytes; the expansion of T-regulatory cells; the production of bioactive compounds against lung pathogens; inhibition of virulence factors; and increase in the phagocytic activity of alveolar macrophages [130,131,132]. Another study found that *L. reuteri* could help to improve the dysbiosis of the CF gut microbiota, which is marked by the high density of proteobacterial species. In addition, there was a considerable reduction in intestinal inflammation, as well as a decrease in the levels of the gut inflammatory marker “calprotectin” and an increase in microbial diversity [133]. Thus, these studies support the view that certain *L. reuteri* strains influence the immune responses in the GI tract.

## 9. Resume of Activities and Global Market for *Limosilactobacillus reuteri*

The market for probiotic foods and supplements is growing fast, and the recent pandemic of SarsCov2 has given a further push to this trend. Accordingly, all the potential healthy activities of *L. reuteri* reported in this review and summarized in Table 1, have led to the development of new products in the probiotics market. Firstly, we want to underline that to the best of our knowledge, there are no food products in the market that contain probiotic strains of *L. reuteri*. However, many papers describe the functionalization of foods with *L. reuteri* strains [134,135,136] On the other hand, the market for supplements is full of products containing *L. reuteri* probiotic strains. Here, we want to give the reader an idea of the marketed products that contain this microorganism. One of the most known companies working in the field of probiotics, particularly on *L. reuteri*, is BioGaia. Indeed, this company, headquartered in Sweden, produces 17 supplements containing combinations of three different strains of *L. reuteri*, the most popular being the DSM 17938, the ATCC PTA 5289 and ATCC PTA 6475 strains, with other healthy ingredients. These products are focused on the gut, gastrointestinal, oral and bone health. Another company is the English Optibac Probiotics, which produce a supplement containing the strain RC-14 for vaginal health. The strain PBS072 is an ingredient in Lactobacillus Reuteri, a supplement from the Italian company, Yamamoto Research. An inactivated formula of the strain *L. reuteri* DSM 17648 is produced in Germany by the company formerly known as Naturawerk, now called Sanatura. Finally, the world leading Danish company in the microbial biomass market, Chr Hansen, produces the strains NCIMB 30242 and UALre-16.

## 10. Conclusions

In humans, the abundance of *L. reuteri* has decreased in recent decades, most likely due to modern lifestyles (e.g., antibiotic use and the Western diet). Over the same time span, there has been an increase in the incidence of inflammatory disorders. This review strongly suggests that *L. reuteri* and its metabolites promote human health through diverse mechanisms. In Table 1, we have listed the probiotic strains mentioned in this paper and their effects. They are effective in decreasing obesity parameters, improving insulin sensitivity and glucose homeostasis, modulating gut integrity, improving inflammation, attenuating hepatic disorders and help to improve the dysbiosis of CF patients. However, there is a need for further investigation on different *L. reuteri* metabolites and strains, especially in humans. In addition, many of the probiotic effects of *L. reuteri* are strain-dependent, as there are many strains with different host origins. As a result, combining several strains of *L. reuteri* to enhance their positive benefits may be useful. Most papers that support the beneficial effects of *L. reuteri* on humans are observational studies and only a few of them define the correlation between its metabolism and outcomes. As is well known, major national and international authorities such as the EFSA and FDA, have not authorized the health claims made for probiotics, although the generally beneficial effects of probiotics on the gut microbiota are fully recognized. This is mainly due to the lack of a clear correlation between probiotic metabolism and the specific beneficial effect. Accordingly, it is pivotal to investigate the metabolism of probiotics and to identify their precise correlation with healthy effects.

## Figures and Tables

**Figure 1 microorganisms-10-00522-f001:**
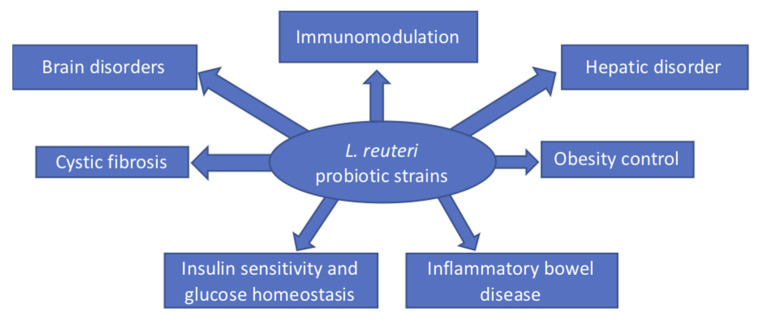
Involvement of *L. reuteri* in different human pathological conditions.

**Figure 2 microorganisms-10-00522-f002:**
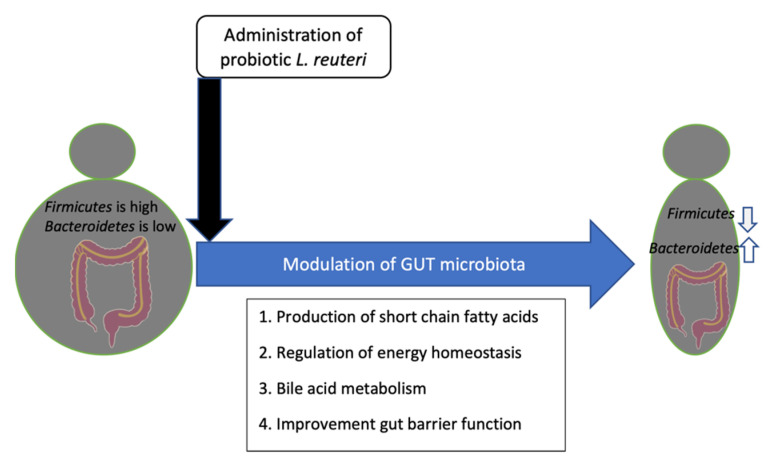
Effect of probiotic *L. reuteri* strains on obese people.

**Figure 3 microorganisms-10-00522-f003:**
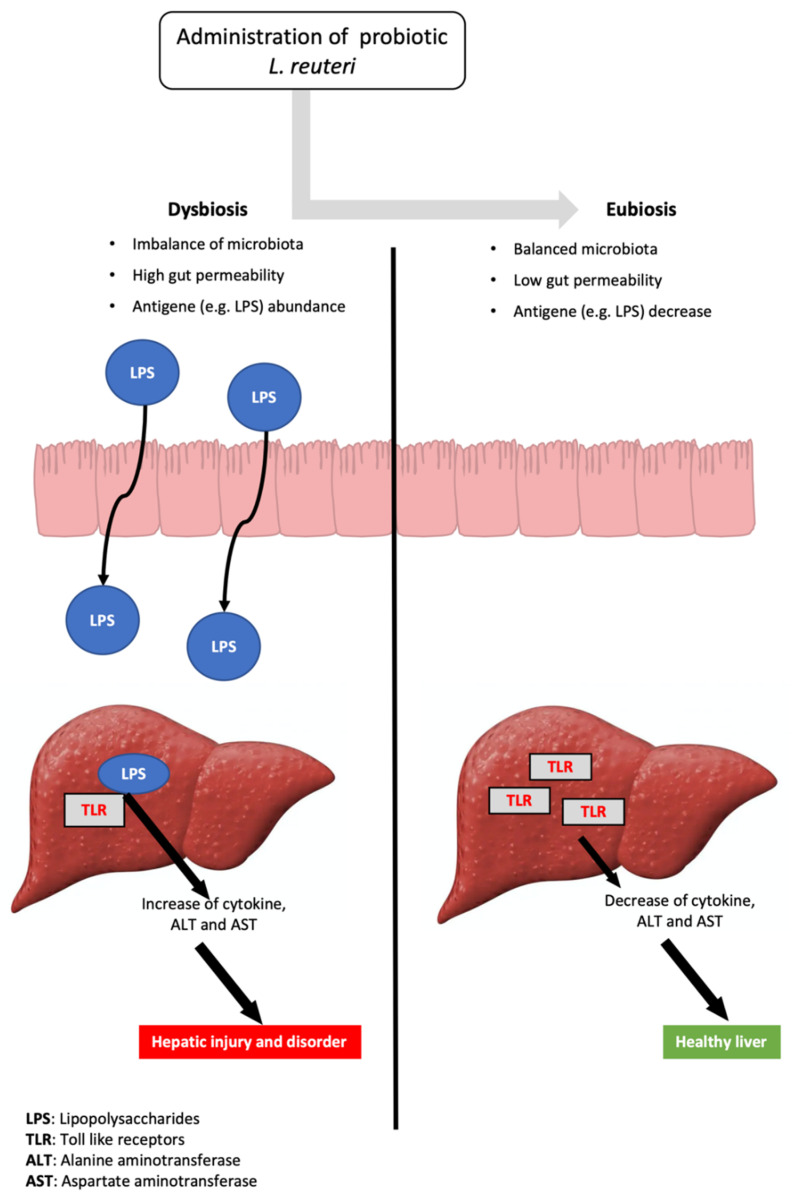
Administration of probiotic *L. reuteri* strains can alleviate hepatic injury and disorder.

**Table 1 microorganisms-10-00522-t001:** Strains of *L. reuteri* cited in this review and their outcomes.

Strain	Outcomes	Reference
MG5149	reduced body weight, adipose tissue weight, and adipocyte size in epididymal tissue. inhibited the expression of lipogenic proteins. reduced fat accumulation.	[26]
JBD301	lowered the concentration of free fatty acids (FFAs) in the small intestine’s gut fluid content. lowered intestinal FFA absorption increased fecal FFA excretion.	[27]
SD5865	increased glucose-stimulated GLP-1 and GLP-2 improved insulin sensitivity. elevated insulin secretion through increasing incretin release.	[32]
GMNL-263	decreased serum glucose, insulin resistance, leptin, C-peptide, and GLP-1. reduced HbA1c level. decreased the load of Clostridia and Bacteroidetes while boosting the number of bifidobacteria and lactobacilli. lowered the aspartate aminotransferase (AST) and alanine aminotransferase (ALT) levels and the antioxidant activities of SOD and glutathione reductase. reduced hepatic inflammation and apoptosis	[36,89,90]
GMNL-89	reduced hepatic inflammation and apoptosis	[90]
DSM 17938	reduced gamma-glutamyl transferase, serum alanine aminotransferase, aspartate aminotransferase, IL-1, IL-2, IL-18, M-CSF, and MIP-3 levels, reduced tissue disorders in the liver and last part of ileum. decreased the abundance of some potential pathogenic taxa. lowered the production of inflammatory genes in the liver. downregulation of retinol metabolism and the peroxisome proliferator-activated receptor (PPAR) signalling pathway. elevation of viral protein association with cytokine and cytokine receptor, and central carbon metabolism in the cancer signalling pathway.	[36,86,87,88]
ZJ617s	diminished gut permeability suppressed the secretion of hepatic pro-inflammatory cytokines. decreased plasma concentrations of AST and ALT. alleviated hepatic histological changes.	[80]
CCM 3625	decrease the inflammatory response in the gastrointestinal tract.	[60]
R2LC	decreased inflammation of the intestinal mucosa	[113]
4659	decreased inflammation of the intestinal mucosa	[113]
F-9-35	has anti-inflammatory benefits	[114]
ATCC55730	lowering the incidence of pulmonary exacerbations and the number of upper respiratory tract infections.	[127]

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
