# Peer review of "Limosilactobacillus reuteri in Health and Disease"

_microorganisms, 2022, doi:10.3390/microorganisms10030522_

Round 1
Reviewer 1 Report
The authors reviewed the probiotic, antimicrobial, and immunomodulatory effects of L. reuteri strains related to their metabolism for amelioration of many diseases and disorders. The work is well organized, the language is good and the outcome is quite interesting.
However, I strongly believe that the following changes should be made:
- I do not feel that the title is representative. Do the authors believe that L. reuteri strains are the best regarding the properties that were addressed? I believe that the title should modified in a way to highlight the indeed good presented properties of this bacteria
- I advise the authors to prepare a Table regarding the sources that L. reuteri strains have been isolated or/and a Table presenting some probiotic L. reuteri strains.
- The authors could present a paragraph (or Table) regarding the application of probiotic L. reuteri strains by Food Industry in Industrial scale (if there are these data). If these data exist, then their presentation is out of the question.
- Regarding the useful data presented regarding obesity, glucose homeostasis, immunomodulation etc, the authors should check again and present wherever it needs, separately or more obviously the data obtained through human trials, besides in vitro and in vivo tests.
- Do the bacteriocins produced by L. reuteri strains have applications by the Food Industry? Please reply.
Author Response
Q. I do not feel that the title is representative. Do the authors believe that L. reuteri strains are the best regarding the properties that were addressed? I believe that the title should modified in a way to highlight the indeed good presented properties of this bacteria
R. Honestly we don’t think title of a manuscript, any type of manuscript, should be “representative” of what is reported in the manuscript. We reckon title should be explicative, attractive and intriguing, with the final purpose to increase the chance the manuscript is read, with benefits to authors and editor. We choose the title like a pun on the common saying “dog is the human’s best friend”. We didn’t believe L. reuteri strains are definitively the best probiotic microorgsnisms, of course. Even because we don’t believe the best probiotic strain exists. Anyway, we would like to maintain the same title but we propose a new title too: Limosilactobacillus reuteri in health and disease.
Q. I advise the authors to prepare a Table regarding the sources that L. reuteri strains have been isolated or/and a Table presenting some probiotic L. reuteri strains.
R. Thank you for you useful suggestion. We add a table as you suggested, it’s the Table 1 in the Conclusion section.
Q. The authors could present a paragraph (or Table) regarding the application of probiotic L. reuteri strains by Food Industry in Industrial scale (if there are these data). If these data exist, then their presentation is out of the question.
R. We focused on the human health and disease away from the food application of L. reuteri strains. Indeed, we don’t distinguish food or supplement use of the microorganism. It’s obvious that use in supplement is much more big than use in food and even though we want to support the food probiotication this is not the paper to do that.
Q. Regarding the useful data presented regarding obesity, glucose homeostasis, immunomodulation etc, the authors should check again and present wherever it needs, separately or more obviously the data obtained through human trials, besides in vitro and in vivo tests.
R. We reckon that this separation could be too much systematic for a generic and a not systematic review and annoying for the reader. In fact, we decided from the beginning don’t write a systematic review to be freer in the organization of the paper and to be more conversational in the style. I hope this is not controversial for you.
Q. Do the bacteriocins produced by L. reuteri strains have applications by the Food Industry? Please reply.
R. We want to clarify here that the antimicrobial activity of L. reuteri is not related to the bacteriocin production but mainly to the reuterin production, that is not a bacteriocin. There are some papers describing reuterin application in foods but at the best of our knowledge its use is not allowed in the food industry.
Reviewer 2 Report
The manuscript entitled “Human’s best friend … Limosilactobacillus reuteri”, describes a good and interesting review regarding the use of this microorganism in different disorders and diseases. In addition, the significant quality of the article is the effect of L. reuteri on obesity, T2D, and several other genetic or gastrointestinal disorders.
But in the text separate and evidence studies according to whether they are done in animals in humans or in vitro.
The review could be also a little bit more critical, and explain every statement and mechanism underlying those statements.
For further improvement, the following minor revisions are required:
- First of all, the manuscript has to be revised by a Native English speaker, as there are several orthographical and spelling mistakes.
- line 10 -12 – please rephrase, it is hard to understand
- lines 31-40 – this section needs more references i.e.: https://doi.org/10.1016/j.idairyj.2021.104997, https://doi.org/10.1016/j.fm.2016.08.009, https://doi.org/10.3390/foods9121894
- in every subtitle the microorganism name has to be put in italics, also revise this aspect in the text, title, and in the references section also (i.e. line 72, 187, 188, 247, 300, 342, 660, 673, etc.).
- line 84 – “gut dysbiosis” - Maybe the definition from line 158 should be moved at the first mention of this term, and also the term dysbiosis is a simplification and can be used as a mental shortcut. The authors should make readers aware of the individual and complex nature of the gut microbiota. It is important to emphasize that there is no "healthy" microbiota.
- line 90 – please insert the term people after obese
- line 99 – please correct stimulus to stimulate
- line 105 – please correct reuteri is the cause to L. reuteri can be the cause to...
- line 143 – please correct this to which
- line 146 – please correct such to such as
- line 148 please correct “toxins are inactivated” to “toxin inactivation”
- line 217 – please correct – tratment
- line 219 – please define “art”
- line 234 – please define the abbreviation (SOD), the same at line 266 (SCFA), 280 (cfu)
- line 268 – the term prebiotic was not defined only mentioned in the manuscript. Please define according to these recent researches: https://doi.org/10.3390/ijerph19031208, https://doi.org/10.3390/nu13062112
- line 273-274 – has to be based on some research i.e. https://doi.org/10.3390/foods10102275
- line 275 – the term autism and abbreviation was already mentioned at line 255
- line 328 - The reference is missing for Marco Colonna et al.
- line 348 – correct popilation
- line 388 - besides antibiotic use, none of the other items from the bracket have been mentioned in the article. please mention these items also above.
- line 398 – 400 – please correct the sentence
Overall, the manuscript is well written with significant data which is sufficiently described and discussed. After major revisions, it could be considered for publication.
Author Response
Q. But in the text separate and evidence studies according to whether they are done in animals in humans or in vitro.
R. We reckon that this separation could be too much systematic for a generic and a not systematic review and annoying for the reader. In fact, we decided from the beginning don’t write a systematic review to be freer in the organization of the paper and to be more conversational in the style. I hope this is not controversial for you.
Q. The review could be also a little bit more critical, and explain every statement and mechanism underlying those statements.
R. Thank you for your suggestion, we added some sentences you find highlighted in the revised version of the paper.
Q. First of all, the manuscript has to be revised by a Native English speaker, as there are several orthographical and spelling mistakes.
R. We revised the English of the text and submit for a last revision by a native English speaker.
Q. Line 10 -12 – please rephrase, it is hard to understand
R. We rephrased the sentence.
Q. Lines 31-40 – this section needs more references.
R. We added
Q. In every subtitle the microorganism name has to be put in italics, also revise this aspect in the text, title, and in the references section also (i.e. line 72, 187, 188, 247, 300, 342, 660, 673, etc.).
R. Thank you for your comment. However, we wrote in italic the name of microorganisms in the subtitles and title, but it seems processing of journal web site changed the character. The name of microorganisms in the references should be automatically changed by the journal editing. We don’t have to take care of reference style, just the DOI is important, according to the journal instruction.
Q. Line 84 – “gut dysbiosis” - Maybe the definition from line 158 should be moved at the first mention of this term, and also the term dysbiosis is a simplification and can be used as a mental shortcut. The authors should make readers aware of the individual and complex nature of the gut microbiota. It is important to emphasize that there is no "healthy" microbiota.
R. We moved the definition of “gut dysbiosis”. You are totally right, there is no “healthy” microbiota. We added a sentence to clarify the concept.
Q. line 90 – please insert the term people after obese
R. Done
Q. line 99 – please correct stimulus to stimulate
R. Done
Q. line 105 – please correct reuteri is the cause to L. reuteri can be the cause to...
R. Done
Q. line 143 – please correct this to which
R. Done
Q. line 146 – please correct such to such as
R. Done
Q. line 148 please correct “toxins are inactivated” to “toxin inactivation”
R. Done
Q. line 217 – please correct – treatment
R. Done
Q. line 219 – please define “art”
R. It is “part”
Q. line 234 – please define the abbreviation (SOD), the same at line 266 (SCFA), 280 (cfu)
R. We fixed all abbreviations
Q. line 268 – the term prebiotic was not defined only mentioned in the manuscript. Please define according to these recent researches: https://doi.org/10.3390/ijerph19031208, https://doi.org/10.3390/nu13062112
R. We added the definition in the Introduction and cited the work of Precup et al.
Q. line 273-274 – has to be based on some research i.e. https://doi.org/10.3390/foods10102275
R. We added the suggested reference.
Q. line 275 – the term autism and abbreviation was already mentioned at line 255
R. Correct, we changed accordingly.
Q. line 328 - The reference is missing for Marco Colonna et al.
R. The correct reference is Cervantes-Barragan et al., Marco Colonna is not the first author, we changed accordingly.
Q. line 348 – correct popilation
R. Done
Q. line 388 - besides antibiotic use, none of the other items from the bracket have been mentioned in the article. please mention these items also above.
R. We changed accordingly.
Q. line 398 – 400 – please correct the sentence
R. We changed the sentence.
Round 2
Reviewer 1 Report
Most of my queries were covered. I still believe that data should be added in the text, regarding commercialized product containing L. reuteri. I insist on that. I would like an answer by the authors it these data exist in the Market in any form and somehow their answer to be added in the text in a sentence.
Author Response
Q. I still believe that data should be added in the text, regarding commercialized product containing L. reuteri. I insist on that. I would like an answer by the authors it these data exist in the Market in any form and somehow their answer to be added in the text in a sentence.
R. Thank you to stimulate us to add this part in the manuscript. We added a new section entitled Resume of activities and global market of Limosilactobacillus reuteri. I hope it's fine for you.
Reviewer 2 Report
The authors improved significantly the manuscript.
Although the English still seems rather hard, there are some parts that should be revised for better understanding.
Some other important corrections:
Line 80 – the authors forgot to introduce a reference – please correct
line 424 – “In the table 1 we list the probiotic strains mentioned in this paper and their effects.” – this sentence is not a conclusion. please move the table from the conclusion.
Otherwise, after revision and important corrections, the manuscript could be published.
Author Response
Q. Although the English still seems rather hard, there are some parts that should be revised for better understanding.
R. Thank you for your comment. Honestly, By a new and more accurate reading we found some sentences not very clear, so we changed to make them more comprehensible.
Q. Line 80 – the authors forgot to introduce a reference – please correct.
R. Done.
Q. line 424 – “In the table 1 we list the probiotic strains mentioned in this paper and their effects.” – this sentence is not a conclusion. please move the table from the conclusion.
R. We moved in a new section.
This manuscript is a resubmission of an earlier submission. The following is a list of the peer review reports and author responses from that submission.